# Optical Method and Biochemical Source for the Assessment of the Middle-Molecule Uremic Toxin β2-Microglobulin in Spent Dialysate

**DOI:** 10.3390/toxins13040255

**Published:** 2021-03-31

**Authors:** Joosep Paats, Annika Adoberg, Jürgen Arund, Ivo Fridolin, Kai Lauri, Liisi Leis, Merike Luman, Risto Tanner

**Affiliations:** 1Department of Health Technologies, Tallinn University of Technology, 19086 Tallinn, Estonia; Jurgen.Arund@taltech.ee (J.A.); Ivo.Fridolin@taltech.ee (I.F.); Kai.Lauri@synlab.ee (K.L.); Merike.Luman@regionaalhaigla.ee (M.L.); Risto.Tanner@taltech.ee (R.T.); 2Centre of Nephrology, North Estonia Medical Centre, 13419 Tallinn, Estonia; Annika.Adoberg@regionaalhaigla.ee (A.A.); Liisi.Leis@regionaalhaigla.ee (L.L.); 3SYNLAB Eesti OÜ, 10138 Tallinn, Estonia

**Keywords:** β2-microglobulin, hemodialysis, dialysis adequacy, middle molecule uremic toxins, optical monitoring, ultraviolet absorbance, fluorescence

## Abstract

Optical monitoring of spent dialysate has been used to estimate the removal of water-soluble low molecular weight as well as protein-bound uremic toxins from the blood of end stage kidney disease (ESKD) patients. The aim of this work was to develop an optical method to estimate the removal of β2-microglobulin (β2M), a marker of middle molecule (MM) uremic toxins, during hemodialysis (HD) treatment. Ultraviolet (UV) and fluorescence spectra of dialysate samples were recorded from 88 dialysis sessions of 22 ESKD patients, receiving four different settings of dialysis treatments. Stepwise regression was used to obtain the best model for the assessment of β2M concentration in the spent dialysate. The correlation coefficient 0.958 and an accuracy of 0.000 ± 0.304 mg/L was achieved between laboratory and optically estimated β2M concentrations in spent dialysate for the entire cohort. Optically and laboratory estimated reduction ratio (RR) and total removed solute (TRS) of β2M were not statistically different (*p* > 0.35). Dialytic elimination of MM uremic toxin β2M can be followed optically during dialysis treatment of ESKD patients. The main contributors to the optical signal of the MM fraction in the spent dialysate were provisionally identified as tryptophan (Trp) in small peptides and proteins, and advanced glycation end-products.

## 1. Introduction

The largest number of the end stage kidney disease (ESKD) patients are treated using hemodialysis (HD), which has remained one of the most expensive and time-consuming methods among the treatments of chronic diseases. Therefore, the monitoring of HD quality, related to the removal efficiency of the uremic solutes in dialysis, is important to ensure adequacy and cost-efficiency of the HD procedure [1,2]. Optical monitoring of the spent dialysate on the outflow from the dialysis machine is a promising alternative to dialysis adequacy estimation based on blood sampling [3]. While ultraviolet (UV) absorbance monitoring of the spent dialysate [4,5,6] enables determining urea-based dialysis quality parameters [7,8], the potential of optical dialysis monitoring is wider. Earlier research has shown the potential of optical monitoring of the spent dialysate to estimate the removal of low molecular weight uremic toxins [9,10,11,12,13,14], elimination of electrolytes, such as phosphate and calcium, [15,16], and possibly for nutrition assessment [17]. As a result, the UV monitoring of the removal of low molecule weight uremic solutes by HD treatment has become commonly applied in the treatment practice worldwide [18,19,20]. Furthermore, fluorescence of spent dialysate has proved to be well applicable for on-line removal monitoring of protein-bound uremic toxins [21,22,23]. However, many recent publications have confirmed the essential role of middle molecule (MM) group of uremic toxins in pathology and mortality of ESKD patients [24,25,26,27,28]. Despite promising works exploring the correlation between optical properties of spent dialysate and MM molecules, the biochemical origin of the MM toxins’ optical contribution to the optical signal has not yet been revealed [29,30,31]. Still, the optimization of the removal of MM uremic solutes has remained an unsolved problem in the treatment of ESKD patients [32].

The aim of this study was to develop an improved optical method based on the UV absorbance and fluorescence of the spent dialysate for the assessment of the concentration of β2-microglobulin (β2M) in the spent dialysate as the specific marker of MM uremic toxins in dialysate, and to explore the methods’ biochemical origins.

## 2. Results

### 2.1. Correlations between Optical Data and Concentration of β2M in Dialysate

Correlations between UV absorbance of spent dialysate at different wavelengths and concentration of β2M in spent dialysate can be seen in Figure 1 based on the data from all three hemodiafiltration (HDF) modalities (including the tank samples). The best correlation was observed at the UV wavelengths around 222 nm (R^2^ = 0.881). The range of UV absorption of aromatic amino acids at 275–280 nm exhibited a slightly weaker correlation (R^2^ = 0.862); surprisingly, quite a good correlation became evident at the wavelength of 311 nm (R^2^ = 0.865).

The best correlation between the fluorescence of spent dialysate and the concentration of β2M in spent dialysate was found in the wavelength region Ex350–370/Em500–555 nm with the coefficient of determination R^2^ up to 0.859 (Figure 2).

The highest correlation between the spectral data of the dialysate and the concentration of β2M was observed in the case of combined UV absorbance with the fluorescence. The best correlations between UV absorbance, the excitation/emission of spent dialysate and the concentration of the β2M in spent dialysate were selected using multiple regression analysis yielding a model including the following optical parameters; UV absorbance at 280 nm, fluorescence Ex280/Em325 and Ex350/Em555 (adjusted R 0.958, accuracy (BIAS ± SE) 0.000 ± 0.304, N = 375).

The identity and Bland Altman plots of the calibration and the validation groups comparing β2M determined at the clinical laboratory (Lab) and predicted optically (Opt), can be seen in Figure 3. The concentration of β2M in the dialysate for both groups was calculated with the algorithm, derived from regression data of the calibration group using optical parameters showing the best correlation combination. The correlation coefficient 0.966 with the accuracy of 0.000 ± 0.272 mg/L was achieved for the calibration group, and the correlation coefficient 0.953 with the accuracy of 0.061 ± 0.340 mg/L was achieved for the validation group between the laboratory estimated β2M concentration in spent dialysate and the corresponding values predicted by the optical model, respectively.

Figure 4 shows average changes of β2M concentration in the spent dialysate during the dialysis treatments estimated in parallel at the laboratory and optically for the validation set. The optically derived β2M concentration time profile corresponds well with the lab data.

Corresponding clinical output for the validation set as the reduction ratio (RR) and total removed solute (TRS) values for β2M are presented in Table 1 using the best multi-regression model (Figure 3) for β2M concentration prediction from the optical measurements. Optically and laboratory estimated values of RR and TRS were not statistically different (*p* > 0.34 and *p* > 0.35, respectively).

### 2.2. UV and Fluorescence Spectra of the MM Fraction

Figure 5 shows the average UV spectra measured as the difference between untreated spent dialysate samples and filtrates, containing solutes < 1kDa of the corresponding dialysate from standard HDF (stHDF) and low flux HD (LF HD) modalities (see Table 4 for dialysis settings). The most noticeable difference in UV absorbance referable to compounds with MW > 1 kDa seems to be in the wavelength region of 210–230 nm. Differences between the spectra of dialysate and corresponding filtrates, referable to compounds with MW > 1 kDa, were considered as characteristics of “MM fraction” hereinafter.

Figure 6 shows the interrelationship between UV absorbance of untreated spent dialysate samples and corresponding filtrates, containing solutes < 1 kDa, from LF HD and stHDF modalities. The very high R^2^ value indicates that variation between optical signals of dialysate samples and corresponding filtrates, caused by the MM fraction, is negligible in the UV region > 230 nm. 

The mean fluorescence emission spectra of MM fractions with characteristic wavelengths of excitation, where the largest difference between MM fractions of LF HD and stHDF can be seen, are shown in Figure 7. Emission wavelengths are presented starting with the lowest meaningful excitation wavelength (i.e., for Ex220 nm emission starts from 230 nm; for Ex280 nm Em starts from 290 nm, and for Ex350 nm Em starts from 360 nm). The predominant maximum emission of the expected MM fraction is evident in the wavelength region Ex280/Em325–335 nm. The fluorescence characteristic to advanced glycation end-products (AGEs) Ex350/Em430–550 nm is present in Figure 7, but with ~10 times weaker intensity compared to fluorescence at Ex280/Em325–335 nm.

For a subset of the stHDF samples collected 7 min after the start (N = 17), MM fraction contributed on average 26.09 ± 6.68% at Ex280/Em325 nm to overall fluorescence of the spent dialysate.

Figure 8 shows that the correlation between the optical signals of dialysate samples and corresponding filtrates is lowest in the wavelength region Ex280/Em320–330 nm due to removal of the MM fraction by filtration. This coincides with the predominant maximum emission region of the MM fraction in Figure 7. The lowest correlation is seen at Ex280/Em325 nm, with the coefficient of determination R^2^ equal to 0.76.

## 3. Discussion

The main findings of the study were: (1) concentration of MM uremic toxin β2M in spent dialysate can be assessed using the UV absorbance and fluorescence of the spent dialysate; (2) β2M concentration prediction from the optical measurements can be used for intradialytic β2M removal assessment as the RR and the TRS; (3) the main contributions to the optical signal of the MM fraction arise apparently from the fluorescence of tryptophan (Trp) in small proteins and peptides and the fluorescence of AGEs, whereas UV absorbance of the peptide bond and aromatic side chains of amino acids seems to have a smaller contribution.

Strong correlation and high accuracy were achieved in comparing the biochemically and optically estimated concentration of the MM uremic toxin β2M in spent dialysate using multiple regression analysis based on the data from all three HDF modalities (R value 0.966 and 0.953, accuracy 0.000 ± 0.272 mg/L and 0.061 ± 0.340 mg/L for the calibration and the validation group, respectively). This enabled visualizing the changing β2M concentration in the spent dialysate during dialysis (Figure 4) and calculating the intradialytic β2M removal by HDF dialysis as RR and TRS from the optical signal (Table 1). The biochemically and optically assessed RR and TRS values were not statistically different (*p* > 0.34 and *p* > 0.35, respectively). Although the removal of β2M is limited by intercompartmental mass transfer, an RR of 68 ± 2% was achieved in an earlier study of 10 patients with HDF in the post-dilution mode, calculated from pre- and post-treatment serum levels of β2M [33]. Our result of an optically estimated RR of 72.06 ± 7.77% is comparable, considering that two modalities besides the standard HDF were aimed to achieve the maximal removal of middle molecules in our settings. A mean total single session removal of β2M 204.9 ± 53.4 mg was observed by Brunati et al., using standard high-flux bicarbonate dialysis [34], which is similar to the value 228.6 ± 83.9 mg, achieved in this work using optical assessment technology combined with the total dialysate collection.

A comparison of both UV absorbance and fluorescence spectra of untreated HDF dialysate with corresponding filtrates, containing solutes < 1kDa (Figure 5 and Figure 7), as well as the corresponding comparison of HDF and LF HD spent dialysate strongly supports the idea that optical properties of dialysate provide a potential for on-line monitoring of eliminating not only small and protein-bound molecules, but also middle molecule uremic toxins from the blood of ESKD patients [29,30,31]. The highest absorbance in the short wavelength region of UV light could be expected in the light of present knowledge about the peptide nature and amino acid composition of the MM uremic toxins [35]. The most widely used features of the absorbance spectra of proteins are attributed to chromophores present in the sidechains of aromatic amino acids like Trp around 280 nm (ε ∼ 5600 M^−1^ cm^−1^), tyrosine around 275 nm (ε ∼ 1420 M^−1^ cm^−1^) and phenylalanine around 257 nm (ε ∼ 197 M^−1^ cm^−1^) [36]. Our results from LF and stHDF modalities (Figure 6) confirm that the MM fraction does not have considerable contribution to the overall UV absorbance of spent dialysate in relation to other chromophores in spent dialysate [12] at corresponding wavelengths. The peptide bond in proteins has a strong absorbance at around 190 nm (ε ∼ 7000 M^−1^ cm^−1^) and a weaker absorbance between 210 and 220 nm (ε ∼ 100 M^−1^ cm^−1^) [37]. As dialysate has a very high absorbance below 230 nm [38,39], and upper absorbance measurement limit of a spectrophotometer was exceeded for some of the samples, which caused noisy data below 230 nm; the cuvette with a shorter optical path length or dilution of spent dialysate samples must be used for a more detailed search of usability of this UV absorbance region for the assessment of MM removal by dialysis as it enables distinguishing the variation caused by the noise and MM fraction more clearly. 

The abovementioned three amino acids are mainly responsible for the inherent fluorescence properties of proteins, with Trp being the dominant intrinsic fluorophore in proteins (Table 2). The deep negative emission of the peak at Ex220/Em370–380 nm (Figure 7) is evidently caused by excitation light absorbance in the short UV wavelength region (so-called inner filter effect) observed also in many other mixtures of fluoro- and chromophores in solutions [40,41].

In addition to the intrinsic fluorescence of these amino acids, a set of UV-absorbing and fluorescent post-translational modifications, including AGEs, has been described, where free amino groups of aliphatic amino acids are involved [43,44,45,46]. Such types of chromo- and fluorophores are known to accumulate in chronic hemodialysis patients [47,48,49,50] and have a large variety of toxic effects for dialysis patients [51]. Only a small part of AGEs has a fluorescence excitation region in the usual absorption wavelength of proteins [46]. Still, the fluorescence of deproteinized serum, specific to AGEs (Ex370/Em440 nm), has been found to predict mortality in hemodialysis patients [52]. AGE pentosidine, which is a crosslink between arginine and lysine residues in proteins [53], has an absorption at 335 nm and a maximum emission at about 375–385 nm [54,55]. Among other typical fluorescent AGEs, argpyrimidine has high absorbance between 320 and 335 nm, and a fluorescence emission maximum of about 400 nm [56]; the vesperlysine A, a crosslink between two lysine residues, has a maxima of absorption at 302 and 263 nm and maxima of emission at 380 and 440 nm [57]. The crossline, another crosslink between two lysine residues, has an emission maximum at 440 nm with excitation at 380 nm [56]. The human β2M modified by glycation is the major component of hemodialysis-associated amyloidosis and has an emission maximum at 450 nm with an excitation at 360 nm [58]. This coincides with the higher emission intensity above 400 nm of an MM fraction at an excitation of 350 nm measured in the spent dialysate from HDF (Figure 7). Moreover, the fluorescence in the same Ex/Em wavelength region exhibited a high degree of correlation with the concentration of β2M in our experiments (Figure 2), with the highest correlation at even higher emission wavelengths of up to 555 nm. This agreement indicates that the fluorescence of AGEs may be the main source of fluorescence of the MM fraction of spent dialysate at least in the longer excitation wavelength region, where aromatic amino acids of MM fraction do not exhibit fluorescent properties. An emission maximum around 450 nm as well as a clear difference between HD and HDF dialysate in this region (Figure 7) supports the essential role of AGEs in the formation of the fluorescence of the MM fraction. Some shifts in highest correlation values between β2M and emission toward longer wavelengths above 450 nm (Figure 2) may be explained to some degree by the energy transfer between different fluorophores in the dialysate (e.g., Förster resonance energy transfer) [59,60] as well as by re-absorbance by different fluorophores and re-emission in longer wavelength regions.

Some weak emissions of non-modified β2M can be achieved with excitation at 220 nm, but such emissions of spent dialysate were found to be independent of the concentration of β2M in the dialysate leading to the conclusion that the β2M in spent dialysate cannot be directly monitored by spectrophotometric measurements [61]. Our spectra of the MM fraction of dialysate (Figure 7) confirm this statement regarding that neither the emission of glycated nor normal β_2_M are evidently dominant in the total fluorescence of spent dialysate at shorter excitation wavelengths when not corrected for the primary inner filter effect. However, correction of the inner filter effect is crucial to achieve a linear relationship between the fluorophore’s concentration and fluorescence in highly absorbing samples [40,41] such as spent dialysate [38,39]. 

The most essential role in the formation of fluorescence of MM fraction seems to originate from the emission region 320–340 nm, with the average maximum at 332 nm, if excited at 280 nm (Figure 7 and Figure 8), which is common for fluorescence of Trp residues in hydrophobic environment of peptides and proteins [62]. This is also close to intrinsic Trp fluorescence of β2M under native condition with the peak at ~337 nm [63]. MM fractions contributed on average 26.09 ± 6.68% to the overall fluorescence of spent dialysate at Ex280/Em325 nm at the beginning of dialysis from the stHDF modality. Interestingly, for the same dataset, the correlation between optical signals of dialysate samples and the corresponding filtrates, containing solutes < 1 kDa, was lowest in the wavelength region Ex280/Em320–330 nm due to the removal of the MM fraction (R^2^ = 0.76), which confirms the high contribution of the MM fraction to the overall fluorescence signal of spent dialysate. In addition, a good correlation was found between the emission intensity at Ex280/Em325 nm, corrected for the inner filtering effect, and the concentration of β2M in spent dialysate from different HDF modalities and timepoints (R^2^ = 0.838, N = 375). 

Apart from aromatic amino acids, pyridoxin and typical enzyme cofactors such as pyridoxal phosphate, pyridoxamine phosphate, nicotinamide adenine dinucleotide and flavin adenine dinucleotide are listed as the main natural intrinsic fluorophores in living tissues [42], which evidently may be adsorbed to proteins with the dimensions of “middle molecules”.

In future, specific MM uremic toxins behind the optical properties of MM fraction could be identified by separating the MM fraction to individual chromo- and fluorophores. This could provide additional information about differences between patients and validate the method more extensively.

## 4. Conclusions

The present work suggests that multicomponent regression analysis proved to be a useful tool for the combination of absorbance and fluorescence at different wavelength regions for concentration and elimination assessment of the MM uremic toxin β2M during dialysis treatment. Including more independent variables (e.g., patient- and diagnosis-specific) into the multiparameter regression analysis may be the next step. The main contributors in the formation of optical properties of the MM fraction are apparently the fluorescence of Trp in small proteins and peptides and the fluorescence of AGEs; whereas UV absorbance of peptide bond and aromatic side chains of amino acids seems to have smaller contribution. Complicated mutual influences of chromophores and fluorophores in dialysate do not allow to attribute distinct excitation/emission wavelength pairs to specific fluorophores in the complex mixture of chromo- and fluorophores present in dialysate. Some presumptive assessments resulting from the phenomenon of the correlations found in this work allow evaluating the optical technology as promising for on-line optical monitoring of eliminating not only β2M but also all of the MM fraction during dialysis treatment. 

## 5. Materials and Methods

In total, 22 ESKD patients were enrolled into the study from the Centre of Nephrology at the North Estonia Medical Centre, Tallinn, Estonia. The study was approved by the Tallinn Medical Research Ethics Committee in Estonia (decision no. 2205, 27. Dec. 2017) and conducted in accordance with the Declaration of Helsinki. Inclusion criteria were the following: over 18 years old patients on chronic hemodialysis, HD procedures via AV fistula or graft (catheters were not used) for 4 h thrice weekly, blood access capable to manage blood flow of at least 300 mL/min, absence of clinical signs of infection or other active acute clinical complications and an estimated life expectancy over 6 months. Clinical data of the participants monitored for a total of 88 HD sessions is presented in Table 3.

Each patient was observed during four midweek dialysis sessions, using Fresenius 5008 dialysis machines (Fresenius Medical Care, Bad Homburg v. d. Höfe, Germany). In order to include large variety of treatment settings, the following dialysis modalities, dialyzers and machine settings were applied (Table 4): (1) standard HDF dialysis with standard settings (stHDF), as previously prescribed for the patient in routine clinical care. This provided a baseline and introduced patients more smoothly into the study; (2) low flux HD (LF HD) with minimal dialysis settings to provide conditions for minimal uremic toxin removal; (3) medium HDF with maximum dialyzer surface area and highest dialysate (d) blood (b) flow ratio (Qd/Qb); (4) high HDF with maximum dialysis settings in terms of dialyzer surface area, dialysate and blood flow, and the substitution volume expected to increase removal of MM. 

Spent dialysate samples were taken from the dialysate outlet of the dialysis machine before dialysis, at 7, 60, 120 and 180 min after the start of the session and at the end of the session (240 min). In addition, the waste dialysate was collected into a large tank during the whole procedure. After the end of the procedure, the dialysate collection tank was weighed, and one sample was taken from it after careful stirring. All dialysate samples were divided into three sets: the first set of samples were directly sent to a clinical laboratory for analysis of β2M; another two sets of samples were separated for the analytical laboratory analyses. Prior to transportation, the samples were processed as follows: (A) spent dialysate samples for the clinical laboratory were collected into 120 mL Becton Dickinson Vacutainer urine collection cups (Franklin Lakes, NJ, USA) and drawn thereafter into the Becton Dickinson Vacutainer SST II Advanced 5 mL (Franklin Lakes, NJ, USA) vacuum tubes; (B) one set of samples for analytical chemistry analysis were transported into the lab in 120 mL Becton Dickinson Vacutainer urine collection cups; (C) the third set of samples were subjected to centrifugation. 12 mL of dialysate were loaded into the centrifugal filter tube with MW cut-off limit 1 kDa (Pall Laboratory Macrosep^®^ type MAP001C37, Pal Corp., Ann Arbor, MI, USA) and centrifuged at 37 °C for 40 min with the speed of 5000 rpm (2375.75× *g*, type 2–16 KL centrifuge with the rotor 11,192 from the Sigma Laborzentrifugen GmbH, Osterode am Harz, Germany). Filtrates were pipetted into standard laboratory vials with Teflon-tightened screw caps and transported into the lab together with the non-filtered samples.

The set A of samples was analyzed for β2M by the clinical laboratory (SYNLAB Eesti OÜ, Tallinn, Estonia) using standard sandwich type immunochemical system “Immulite 2000 Beta-2 Microglobulin” (Siemens Healthineers AG, Erlangen, Germany). Sets B and C of samples were subjected to optical analyses during the day of sampling at room temperature, as follows: UV spectra were recorded with the UV-3600 spectrophotometer (Shimadzu, Kyoto, Japan) in the wavelength range of 190–400 nm with the increment of 1 nm using a quartz cuvette with optical path length of 10 mm. An untreated pure dialysis buffer, sampled from the outflow of the dialysis machine prior to switching on the blood flow, was used as the reference solution in UV measurements (separately for each dialysis session) or the filtrate of the dialysate sample was used as the reference for measurement of dialysate for the same patient. Fluorescence spectra were recorded with the spectrofluorometer RF-6000 (Shimadzu, Kyoto, Japan) using the excitation wavelength range of 200–400 nm with the increment of 10 nm and the emission wavelength range of 210–600 nm with the increment of 1 nm. The bandwidths of 5 nm were used in both monochromators and the used quartz cuvette had an optical path length of 4 mm. Differences between the spectra of the dialysate and corresponding filtrates, referable to compounds with MW > 1 kDa, were considered as characteristics of the “MM fraction” and their possible biochemical origin was examined based on data available in the literature. Additionally, a regression analysis was used to analyze the variation of optical signals at different wavelengths, caused by the MM fraction, when comparing the spectra of dialysate samples and corresponding filtrates. Correction of emission intensity in relation to inner filtering of the excitation beam was used in the case of excitation at 280 nm when included into the final regression model [40,41].

Forward stepwise regression was used to obtain regression models for the assessment of the β2M concentration through optical parameters of spent dialysate. Independent variables included UV absorbance and fluorescence with selected excitation/emission wavelength pairs as specified below. The final choice of the best combination of optical parameters was validated by dividing HDF data into a calibration set (50% of the material, i.e., 11 patients with even number of registration, 33 HDF sessions) and a validation set (50% of the material, 11 patients with odd number of registration, 33 HDF sessions).

Systematic error (BIAS) was calculated for the models as follows:(1)$$\mathrm{BIAS}=\frac{\sum_{i=1}^{N}e_{i}}{N}$$
where *e_i_* is the *i*-th residual (difference between the lab and modelled β2M) and N is the number of observations [64].

Standard error of performance corrected for BIAS was calculated for the two models as follows [64].
(2)$$\mathrm{SE}=\sqrt{\frac{\sum_{i=1}^{N}{\left(e_{i}-\mathrm{BIAS}\right)}^{2}}{N-1}}$$

Removal of β2M during dialysis sessions was described by a reduction ratio % (RR) in dialysate:(3)$$\mathrm{RR}=\frac{C_{0}\;-\;C_{t}}{C{\;}_{0}}\times 100\%$$
where C_0_ is the concentration of β2M in a spent dialysate sample taken after 7 min from the start of the dialysis procedure and C_t_ is the concentration of β2M in a spent dialysate sample taken at the end of the dialysis procedure. 

The total removed solute (TRS) of β2M was calculated based on the total dialysate collection (TDC) as follows:(4)$$\;\mathrm{TRS}={\;C}_{T}\times W_{T}$$
where C_T_ is the final substance concentration in the dialysate collection tank and W_T_ is the weight of the dialysate collection tank [kg]. It was assumed that the average density of spent dialysate is 1.008 ± 0.001 kg/L [65]. Both RR and TRS were calculated based on β2M concentration in dialysate estimated in the laboratory as well as by the best optical multiparameter model. Excel MS 365 (Microsoft, Redmond, WA, USA) and MATLAB R2019b (MathWorks, Natick, MA, USA) software were used for statistical analyses. All results were assessed for possible errors and data conformity. Twenty-one data points from the total of 396 were omitted from the data set due to errors related to sample drawing, clinical laboratory analysis and due to self-tests of the HD machine. Individual differences in the laboratory and optically estimated results were examined using a Bland and Altman analysis [66], and a parametric paired t-test (two-tailed) was used to determine the statistical difference between laboratory and optical methods. A *p*-value of <0.05 was considered statistically significant.

## Figures and Tables

**Figure 1 toxins-13-00255-f001:**
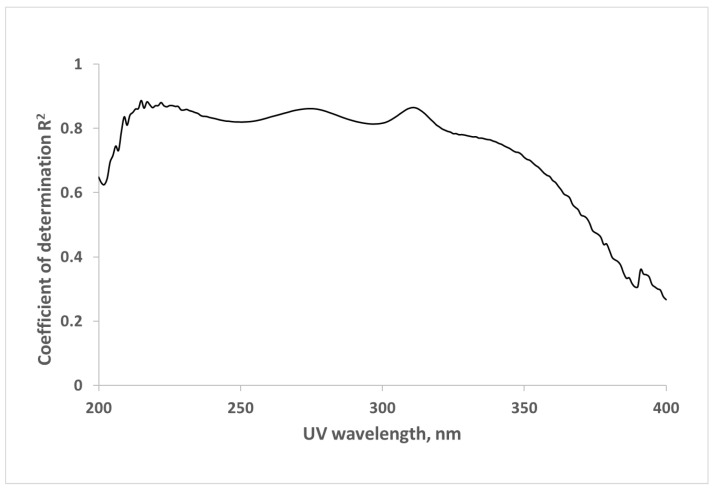
Wavelength dependence of the correlation between ultraviolet (UV)-absorbance of spent dialysate and concentration of β2-microglobulin (β2M) in spent dialysate for hemodiafiltration (HDF) modalities (N = 375).

**Figure 2 toxins-13-00255-f002:**
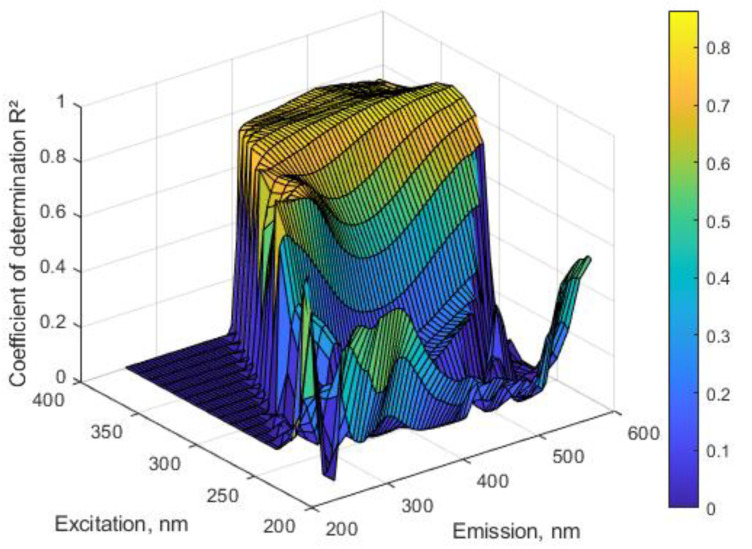
Wavelength dependence of the correlation between fluorescence intensity and concentration of β2M in spent dialysate for HDF modalities (N = 375).

**Figure 3 toxins-13-00255-f003:**
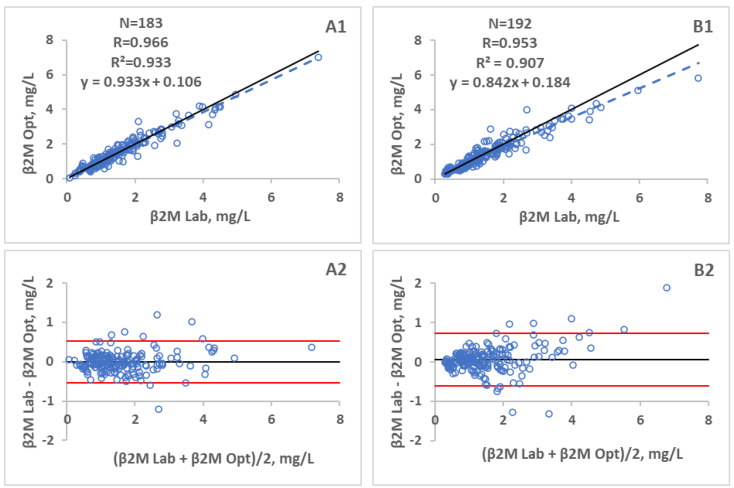
Identity plots of the β2M concentration in spent dialysate determined at the clinical laboratory (Lab) and predicted optically (Opt), and Bland-Altman plots of the differences between Lab and Opt concentrations. **A**—Calibration set, **B**—Validation set, 1—Regression plot, 2—Bland-Altman plot.

**Figure 4 toxins-13-00255-f004:**
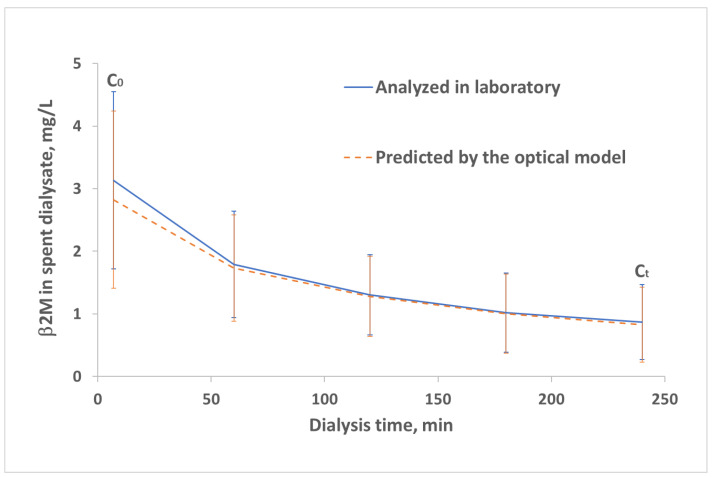
Time-series of changing β2M concentration (mean ± SD) in the spent dialysate during HDF dialysis sessions (N = 29) for patients of the validation set.

**Figure 5 toxins-13-00255-f005:**
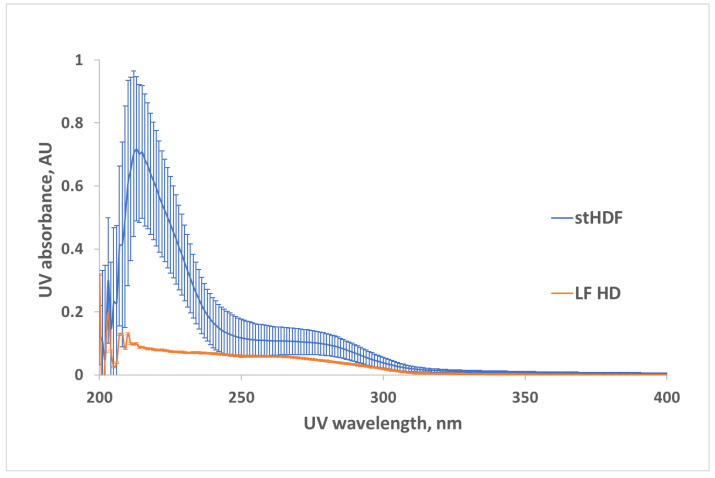
UV spectra (mean ± SD, N = 9) of middle molecule (MM) fractions of 7 min dialysate from low flux HD (LF HD) and standard HDF (stHDF) modality. The untreated dialysate was measured against corresponding filtrate containing solutes < 1 kDa for a subset of samples.

**Figure 6 toxins-13-00255-f006:**
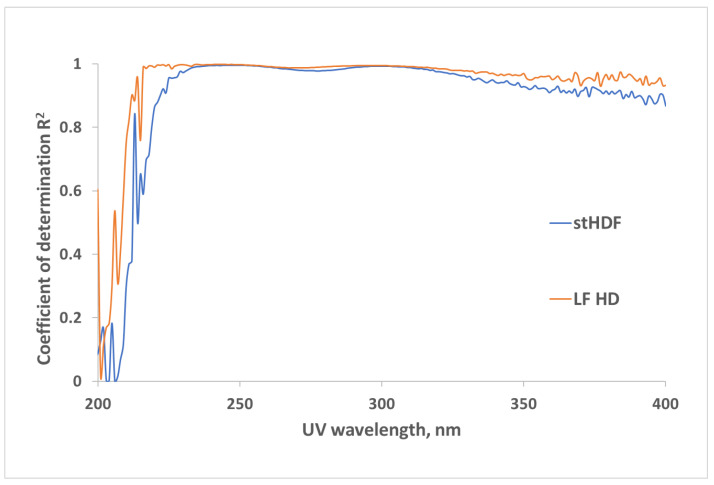
Correlation between UV-absorbance of untreated spent dialysate samples and UV-absorbance of corresponding filtrates, containing solutes < 1 kDa, from LF HD and the stHDF modalities sampled 7 min after the start of the dialysis from a subset of samples (N = 17).

**Figure 7 toxins-13-00255-f007:**
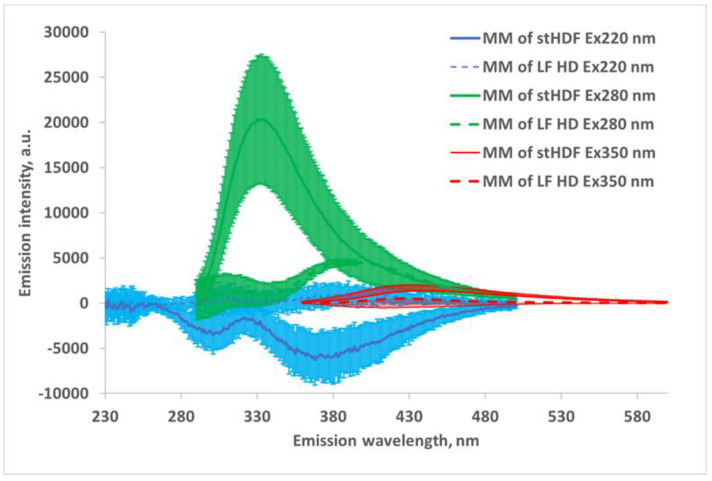
Fluorescence emission spectra (mean ± SD, N = 17) of MM fractions of dialysate measured as the difference between untreated dialysate and corresponding filtrates, containing solutes < 1 kDa, from LF HD and the stHDF modalities sampled 7 min after the start of the dialysis at some characteristic wavelengths from a subset of samples.

**Figure 8 toxins-13-00255-f008:**
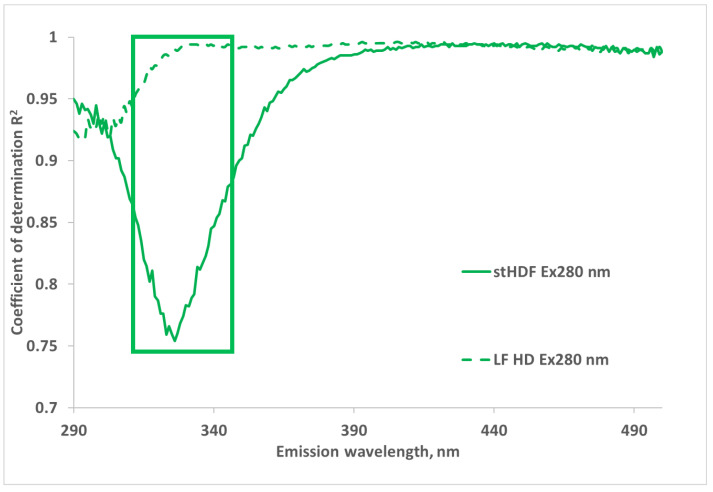
Correlation between fluorescence emission spectra of untreated spent dialysate and corresponding filtrates, containing uremic solutes < 1 kDa, from LF HD and the stHDF modalities sampled 7 min after the start of the dialysis at Ex280 nm from a subset of samples (N = 17). Emission region with the highest correlation between β2M in the spent dialysate, used in the final regression model, is marked with a rectangle.

**Table 1 toxins-13-00255-t001:** Clinical output of β2M removal by HDF dialysis as the average reduction ratio (RR) and the total removed solute (TRS) for the validation set.

Clinical Parameter	β2M Lab mean ± SD	β2M Opt mean ± SD	*p*	Accuracy (BIAS ± SE)	Pearson Correlation
RR (%, N = 31)	73.37 ± 10.39	72.06 ± 7.77	0.35	−1.31 ± 5.41	0.894
TRS (mg, N = 33)	234.5 ± 72.8	228.6 ± 83.9	0.35	−5.95 ± 36.09	0.904

**Table 2 toxins-13-00255-t002:** The fluorescent properties of aromatic amino acids in water at neutral pH [42].

Amino Acid	Excitation Wavelength (nm)	Emission Wavelength (nm)	Bandwidth (nm)	Quantum Yield
Tryptophan	295	353	60	0.13
Tyrosine	275	304	34	0.14
Phenylalanine	260	282	-	0.02

**Table 3 toxins-13-00255-t003:** Clinical data of the end stage kidney disease (ESKD) patients. Numerical values are given as a mean ± SD or as a median and interquartile range (Q1–Q3).

Entity of the Data	Specification
Cause of ESKD	Diabetes (4); Hypertension (8); Glomerulonephritis (3); Tubulointerstitial nephritis (3); Renal carcinoma (2); Other (2)
Age (years)	55 ± 17
Gender	M (17), F (=5)
Race, Caucasian (%)	100
BMI, kg/m^2 a^	26.8 ± 5.8
BW, kg ^a^	81.5 ± 21.3
Ultrafiltration volume, mL	2565 ± 1190
Urinary volume, mL	0 (14 patients) 700 (335–825) (8 patients)
Serum total protein, g/L	62.8 ± 5.5
Hematocrit, % ^a^	34.4 (3.5)
Serum calcium, mmol/L ^a^	2.25 (0.16)
Serum phosphorus, mmol/L ^a^ Serum parathyroid hormone, pmol/L ^a^	1.92 (1.63–2.29) 28.7 (16.8–41.9)
Dialysis access	native fistula (15); graft (7)
Dialysis vintage, months ^a^	23 (11–83)
spKt/Vurea ^a^	1.47 (1.23–1.67)

Abbreviations: M—male; F—female; BMI—body mass index; BW—body weight; spKt/V—single-pool criterion of the dose of dialysis, stHDF—standard HDF. ^a^ Based on data of stHDF.

**Table 4 toxins-13-00255-t004:** Dialysis treatment settings of hemodiafiltration (HDF) and hemodialysis (HD). Numerical values are given as mean ± SD.

Entity of the Data	Standard HDF	Low Flux HD	Medium HDF	High HDF
Volume substituted (Vs, L)	21.1 ± 3.1	0 ± 0	15.3 ± 1.4	25.3 ± 2.8
Dialysis time, min.	240	240	240	240
Blood flow, mL/min (Qb)	300.8 ± 12.7	200 ± 0	299.7 ± 1.0	364.2 ± 27.1
Dialysate flow, ml/min (Qd)	470.8 ± 105.4	300 ± 0	799.8 ± 0.9	800.0 ± 0.0
Dialyzer area ^a^, m^2^	2.0 ± 0.2	1.5 ± 0.0	2.2 ± 0.0	2.2 ± 0.0
Number of dialyses (N)	22	22	22	22

^a^ Specification of dialyzers: Standard: FX800 (N = 8), FX1000 (N = 14), Low flux: Lo15 (N = 22), Medium: FX1000 (N = 22), High: FX1000 (N = 22). All dialyzers had polysulfone-based membranes with the following effective membrane area: 1.8 m^2^ (Helixone^®^, FX800), 2.2 m^2^ (Helixone^®^, FX1000), 1.5 m^2^ (Amembris^®^, Xevonta Lo 15).

## Data Availability

Data sharing is not applicable due to legal and privacy issues.

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
