# Peer review of "Optical Method and Biochemical Source for the Assessment of the Middle-Molecule Uremic Toxin β2-Microglobulin in Spent Dialysate"

_toxins, 2021, doi:10.3390/toxins13040255_

Round 1
Reviewer 1 Report
This paper demonstrate a useful tool for combination of absorbance and fluorescence at different wavelength regions for concentration and elimination assessment of MM uremic toxin β2M during dialysis treatment.
The paper is written with high quality, has a very high relevance.
Author Response
We would like to thank the reviewer for their time and we are thankful for the compliments.
Reviewer 2 Report
Overall, this is interesting study.
The investigators can improve this manuscript by including data on dialysis machine and dialyzers that were used.
There is no data on dialysis catheter as dialysis access?
Instead of total protein, serum albumin should be used
Hemoglobin and data on CKD MBD should be provided
Causes of ESKD should be provided.
How will data from this study apply for the next steps?
Author Response
We would like to thank the reviewer for their time and comments on the manuscript. The manuscript has been edited to address the concerns and we now feel that manuscript improved due to the comments of the reviewer.
Please see the attachment for detailed answers.

Round 2
Reviewer 2 Report
The authors have responded appropriately. This is an interesting paper with increased scientific soundness after their corrections.